# The Associations between the Maxillary Sinus Volume, Infraorbital Ethmoid Cells, and the Infraorbital Canal: A CT-Based Study

**DOI:** 10.3390/diagnostics13233593

**Published:** 2023-12-04

**Authors:** Einat Kedar, Ilan Koren, Bahaa Medlej, Israel Hershkovitz

**Affiliations:** 1Department of Anatomy and Anthropology, Faculty of Medicine, Tel Aviv University, Tel Aviv 6997801, Israel; einatkedar@tauex.tau.ac.il (E.K.); bahaamedlej@gmail.com (B.M.); 2The Dan David Center for Human Evolution and Biohistory Research, Faculty of Medicine, Tel Aviv University, Tel Aviv 6997801, Israel; 3Atid Medical Group, Raphael Hospitals, Tel Aviv 6158101, Israel; ilankorenmd@gmail.com

**Keywords:** maxillary sinus volume, infraorbital ethmoid cells, infraorbital canal, sinusitis, pneumatization

## Abstract

This CT-based study aimed to characterize and explain the existence of two anatomical structures positioned near the maxillary sinuses, which are of clinical relevance in rhinology and maxillofacial surgery. A total of 182 head scans (92 males and 90 females) were inspected for infraorbital ethmoid cells (IECs) and for the type (route) of infraorbital canal (IOC). The maxillary sinuses were segmented, and their volumes were measured. Statistical analysis was conducted to reveal the associations between the two anatomical variations, namely, sex and the maxillary sinus volume. Infraorbital ethmoid cells were noted in 43.9% of the individuals studied; they were more frequent in males (53.3%) than in females (34.4%). The descending infraorbital nerve (type 3 IOC) was found in 13.2% of individuals and was independent of sex. Infraorbital ethmoid cells were associated with the IOC types. The maxillary sinus volume was found to be sex-dependent. A large sinus volume is significantly associated with IOC Type 3 (the descending canal) and the presence of IEC. Dentists, radiologists, and surgeons should be aware that individuals with extensive pneumatization of the maxillary sinuses are more likely to display a descending IOC and IEC. These findings should be studied, along with CT scans, before treatment and surgery.

## 1. Introduction

The nasal cavity and the paranasal sinuses are among the most complex structures in the human body [1]. As such, there are many anatomical variations in this region [2]. Some variations are structural, and many others result from extensive pneumatization. The ethmoid air cells have several variations that interface with the other three paranasal sinuses [2,3]. The most posterior ethmoid cell variation, the Onodi cell (sphenoethmoidal cell), is located superior and lateral to the sphenoid sinus [4], and the most anterior ethmoid air cell, the agger nasi, is located anterior to the frontal recess, next to the frontal sinus opening [3]. The third ethmoid air cell is the infraorbital ethmoid cell (IEC), which is one of the two anatomical structures near the maxillary sinuses examined in this study.

The IEC is located within the inferior-medial wall of the orbital floor lateral to the infundibulum and medial to the infraorbital nerve [5,6]. The cell’s size varies from half a millimeter up to several millimeters; their number can range from none to a few, bilateral or unilateral, and the cells can be adjacent to the bulla (i.e., convoluted bulla) or separated (Haller’s cells) [6,7]. Since they can potentially narrow the sinus opening from within the sinus or its outlet [8], these cells were found to be associated with repeated acute rhinosinusitis [9], with mucosal thickening [10], and with chronic rhinosinusitis in patients with allergic rhinitis [11]. These findings, however, are controversial [5,6,12,13]. Therefore, recognizing the presence of IEC before endoscopic sinus surgery, such as FESS, is clinically important [13].

The second variation of interest is the location of the infraorbital canal (IOC). The IOC is less discussed in the literature despite the infraorbital nerve’s clinical relevance in many medical disciplines, such as ophthalmology, otolaryngology, maxillofacial surgery, and dentistry [14,15]. Usually, the IOC passes through or approaches the roof of the maxillary sinus; however, in about 12% of the cases, the canal can descend into the lumen, which can be dangerous during sinus endoscopic surgery [14,16,17].

Interestingly, a descended IOC is associated with ipsilateral IEC cells [14,17]. This study further examines these relationships with respect to sex and the maxillary sinus volume. Both variations have clinical relevance, and awareness of their presence is essential for surgeons performing surgical procedures on the sinuses [3,18].

## 2. Materials and Methods

A random sample of 207 head CT scans from Carmel Medical Center, Haifa, Israel, was reviewed for the present study. The scans were carried out on a Brilliance 64 or iCT256 scanner, Philips Medical Systems, Cleveland, Ohio; the slice thickness was 0.5–3 mm, the voltage was 120 kV, and the current was 150–570 mA. The scans were examined and analyzed using Phillips IntelliSpace Portal (WIP) 11.2. Twenty-five scans were excluded due to the following: maxillary sinus hypoplasia (*n* = 10), sinus augmentation (*n* = 2), and extensive mucosal thickening (*n* = 13). Individuals with other ethmoid cell variations (e.g., Onodi cell, agger nasi cell) were included in the study. The final sample included 92 males and 90 females, ages 18–58 years (mean age 34.6 and 35.6, respectively).

Each scan was examined on three planes (coronal, sagittal, and axial) to identify the anatomical variations (IEC and IOC). Clinical data regarding sinusitis diagnoses were collected from the medical files (between January 2002 and August 2021) and were based on the ICD-10 code system [19].

### 2.1. Infraorbital Ethmoid Cells

Infraorbital ethmoid cells were identified at the level of the infundibulum (Figure 1). The recorded cells fulfilled the criteria defined by Ahmad et al. [5] and were differentiated from the infraorbital recess of the maxillary sinus.

### 2.2. Infraorbital Canal Location

Three locations of the IOC were recorded following Ference et al. [14] and Haghnegahdar et al. [17]. Type 1 was recorded when the nerve canal was within the bone and outside the maxillary sinus lumen (Figure 2A–C). Type 2 was recorded when the nerve canal was within the roof of the sinus but slightly protruding from the lumen (Figure 2D–F). Type 3 was recorded when the nerve canal fully descended into the sinus lumen (Figure 2G–I).

### 2.3. Evaluating the Maxillary Sinus Volume

The DICOM files’ stack of each scan was imported to Amira software (v. 6.3). The maxillary sinus surfaces (Figure 3) were virtually extracted using semi-automatic and automatic tools. First, to determine the boundaries of the sinuses, the skull was segmented using a threshold tool with masking adjusted to the bone density. Then, in each sinus space, the first, middle, and last slices were marked with the brush tool and interpolated for all slices. Lastly, the sinus surfaces were automatically created using the watershed tool and a gradient image. Unrelated parts (such as the infundibulum) were cleaned from the surface using the lasso tool. The left and right maxillary sinus volumes (L-MSV and R-MSV) were automatically calculated (in cube millimeters) from each sinus surface, using the ‘Surface Area Volume’. An average sinus volume variable was calculated using (RMSV + LMSV)/2.

The maxillary sinus volume was divided into three groups, as follows: “Small”: volume < 12 cm^3^, “Medium”: volume > 12 cm^3^, and <16 cm^3^, and “Large”: volume > 16 cm^3^.

### 2.4. Statistical Analysis

The data structure, statistical analysis, tables, and graphs were created using a specially designed program written by the researcher using the R-4.1.0 software [20] platform and the RStudio version 1.4.1717 editor [21]. Tables were created using the “finalfit” package [22], and the output was created using the “kableExtra” package [23]. Figures were created using the “ggplot2” package [24].

For the reliability of observations, a Kappa test was conducted between the author and a second researcher (the score was set to K > 0.6).

Descriptive statistics were used to obtain data on the prevalence of the two anatomical variations, and a Chi-square test was used to find significant associations between them and to determine their associations with sex. Cochran’s Q test was used to test the IOC type’s side symmetry.

Continuous variables, namely, the right, left, and total maxillary sinus volumes, were tested for normal distribution using the Shapiro–Wilk normality test. A paired sample *t*-test was conducted to examine the symmetry in maxillary sinus volumes. An independent two-sample *t*-test was conducted to locate significant sex differences in the maxillary sinus mean volume.

A two-sample *t*-test was used to determine differences in the sinus mean volume between individuals with and without IEC for each sex separately. Kruskal–Wallis tests were used to determine differences in the sinus mean volume between different types of IOC for each sex separately (the sample size of the type 3 group was *n* < 30).

The Chi-square test was used to find significant associations between each anatomical variation and the recorded sinusitis.

Logistic regression analysis was performed to allocate predictive variables for types of IOC routes and to look for the presence of IEC (the independent variables were sex and sinus volume).

## 3. Results

### 3.1. Prevalence of Infraorbital Ethmoid Cells and Their Association with Sex

Infraorbital ethmoid cells were found in 43.9% of individuals and were sex-dependent, i.e., they were more frequent in males (49 out of 92, 53.3%) than in females (31 out of 90, 34.4%) (χ^2^ = 5.797, df = 1, *p* = 0.016, Table 1). With regard to laterality, no significant differences were found between the sexes (χ^2^ = 2.164, df = 2, *p* = 0.339, Table 1), since in both sexes, the bilateral cells were the most common type (>50%).

### 3.2. Prevalence of the Infraorbital Canal Type and Its Association with Sex

The ICO type was sex-independent (χ^2^ = 1.392, df = 2, *p* = 0.498, Table 2) and side-independent (Cochran’s Q test did not indicate any differences between the left and right, Q = 2.951, df = 1, *p* = 0.085, Table 3). Type 1 was the most common (46.7% on the right and 53.3% on the left side), followed by type 2 (40.1% on the right and 34.1% on the left side). Type 3 was observed in 13.2% (right) and 12.6% (left) of the individuals studied.

### 3.3. Sinus Volume in Males and Females

The mean maxillary sinus volume of the combined sample was similar for the right and left sides (14.7 ± 4.78 cm^3^ for RMSV and 14.6 ± 4.78 cm^3^ for LMSV, *p*-value = 0.382). Therefore, we could use an average sinus volume variable (14.7 ± 4.67 cm^3^) for further analysis.

The average sinus volume was greater in males than in females (15.8 ± 5.04 cm^3^ and 13.5 ± 3.95 cm^3^, respectively, *p*-value < 0.001). Similar significant differences between males and females were found when the mean of the right and left sinus volume were compared separately.

### 3.4. Association between IEC, IOC, and the Sinus Volume

In both males and females, the sinus volume on the left side was significantly greater in individuals with IEC than in those individuals who had none (Figure 4). On the right side, the difference was significant only for males (*p*-value = 0.0087). The logistic regression analysis model with sex and the maxillary sinus volume as independent variables showed that the volume is significantly associated with the presence of IEC; however, contrary to the Chi-square test results, sex is not associated with it. The model showed that an increase in one maxillary sinus volume unit (1 cm^3^) adds 11.5% to the probability of exhibiting IEC (Table 4). Another logistic regression model found that the chance of exhibiting IEC on any side is 3.1 times greater in individuals with a large maxillary sinus volume (volume ≥ 16 cm^3^).

Interestingly, the more protruding the IOC is (from type 1 to type 3), the greater the maxillary sinus volume (Figure 5) and the greater the prevalence of IEC (Table 5). However, the multiple logistic regression model, which tested the associations between type 3 IOC (on the right side only) and the independent variables of sex, IEC, and the maxillary sinus volume, revealed no significant association between the two anatomical variations. Sinus volume, however, is the only factor significantly associated with type 3 IOC; an increase in one maxillary sinus volume unit (1 cm^3^) adds 14.5% to the probability of exhibiting type 3 IOC (Table 6).

It is worthwhile to mention that a single logistic regression model showed very similar results; an increase in one maxillary sinus volume unit (1 cm^3^) adds 15.6% to the probability of exhibiting type 3 IOC. The model of this regression had 86.3% accuracy in predicting the type 3 canal.

### 3.5. Association of IEC and IOC with Sinusitis

Of the 182 individuals studied, 43 (23.6%) had sinusitis (17 males and 26 females, χ^2^ = 2.186, df = 1, *p* = 0.139).

There was no association between the presence of IEC or IOC types and sinusitis (the Chi-square test details are presented in Table 7).

## 4. Discussion

Anatomic variations in the nasal cavity and paranasal sinuses are common and widely investigated [3,5,6,7,8,9,10,16]. Our study revealed an interesting aspect of anatomical variation with respect to the general pneumatization of the skull. We show here that the infraorbital canal type and the infraorbital ethmoid cells are maxillary sinus volume dependent.

### 4.1. Infraorbital Ethmoid Cells

The prevalence of IEC found in this study (43.9%) is within the known range. Previous studies found a prevalence of 38.2% in a panoramic radiograph study, 39.1% in CT scans, and 43.3%, 23.62%, and 60% in CBCT studies [5,6,7,13,25]. The Chi-square test showed that the presence of these cells is sex-dependent (more frequent in males). This is probably because the researchers did not control for sinus volume, which is greater in males, as shown in our study and in others [26,27]. When comparing the frequencies of the presence of cells in males and females separately, the association with the sinus volume remains the same. Logistic regression also shows that the sinus volume is a significant predictive variable for the presence of IEC.

Although IEC is frequently seen, its etiology remains unclear. Furthermore, exactly how pneumatization accrues is unknown [1,28,29]. Currently, the invasive tissue hypothesis is the accepted theory, according to which the epithelial tissue invades the bony structures opportunistically until it is restricted by the surrounding bones [1,29]. Bearing in mind that the IEC belong to the ethmoid sinus, separated from the maxillary sinus, and have their drainage path into the uncinate groove [28], we suggest that their development might be a byproduct of an extensive pneumatization, where the maxillary sinuses and the ethmoid sinuses expand separately, in the same time frame, due to unknown genetic and environmental factors.

Support for this explanation can be found in the embryology stage since both structures evaginate from the uncinate process during the same week [30]. Therefore, their development is directed by the same biological factors. Finally, studies that found associations between the volumes of all the paranasal sinuses and between the overall pneumatization of the skull and the sinus’s volume [31,32,33,34] may indicate that there are common factors that determine the extent of the pneumatization of the skull.

### 4.2. Infraorbital Canal

This study demonstrated, like others, that the most common location of the IOC is within the orbital floor (Type 1) [14,35,36]. However, two other studies reported that Type 2 is the most frequent variation [16,17]. In agreement with all previous studies, the prevalence of Type 3, the descending nerve canal, was the least frequent variation (13.2%) and fell within the known range of 7.9% to 23.2% [14,16,17,35,36]. The symmetry of the IOC type between the right and left sides is also consistent with the literature [36].

Like Ference et al. [14], we also found that IEC correlates with IOC Type 2 and Type 3. However, this association is sinus volume dependent, i.e., it is significant only for large sinuses, implying that the sinus volume is a confounder in this relationship.

Sinus volume was significantly greater in individuals possessing IOC Type 3. To the best of our knowledge, only one study examined the relationships between sinus size and IOC type and reported an association between the two (the evaluation of the maxillary sinus size was based on linear measurements) [36]. Interestingly, these authors found that the sinus’s width and anterior–posterior length correlate with the descending nerve canal, whereas the height does not. They also found a strong relationship between the length of the canal (from the infraorbital groove to the infraorbital foramen) and the length of the sinus, and even stronger relationships between the length of its entire pathway from its origin in the foramen rotundum and the length of the sinus. They concluded that the length of the IOC is cranial size-dependent, receiving support from the study conducted by Kazkayasi et al. [37] that showed that the length of the infraorbital canal is determined by craniofacial size (the distance between the sella turcica and the nasion). Kazkayasi et al. [37] found that as the anterior–posterior lengths (sella turcica to nasion and sella turcica to infraorbital foramen) increase, the foramen rotundum moves forward and downward, which implies that the outlet of the nerve starts lower and closer to the infraorbital foramen. This mechanism ensures that the length of the nerve remains stable.

The route of the IOC is fixed in the late prenatal period when it branches out to innervate the maxillofacial region during the development of the teeth of the upper jaw [38]. Since the maxillary sinus reaches a mature size only at around the age of 16 years [39], and the secondary pneumatization stage actually continues long after the nerve canal is in place, we suggest that as the sinus increases in size, it encompasses the infraorbital nerve canal whose route has already been set. When pneumatization expands more into the sinus roof, the canal appears as it descends into the lumen. Zollikofer and Weissmann demonstrated this process in a computer simulation and have shown how the ‘invasive tissue theory’ works when pneumatization continues around the nerve [29].

### 4.3. Clinical Implications

As noted, the prevalence of IEC in various sinus imaging types is quite high [5,6,7,13,25]. Despite its distinctive location at the maxillary sinus outflow, its potential impact on sinus drainage and its role in promoting acute or chronic sinusitis remain unclear. Some studies found a significant association between the presence of IEC and sinusitis [9,10,11], whereas others have noted that a significant increase in maxillary sinus mucosal disease is associated with the IEC size and not its presence [40,41]. Furthermore, IEC is one of many other factors in the etiology of chronic rhinosinusitis, including allergic rhinitis and asthma [5].

In contrast, our study, like others [5,6,12,42], found no significant relationship between IEC and maxillary sinusitis. Nevertheless, our study, like that of Ference et al. [14], found that the presence of IEC is associated with a descending IOC. The pneumatization process might be the key to understanding these variations since we found that both variations are associated with large maxillary sinuses. However, these relationships also have clinical implications since one of the surgical considerations is the proximity of the IEC to the infraorbital nerve, which runs along the roof of the maxillary sinus, just behind the IEC. Awareness of the IEC’s presence and the course of the IOC before surgery, as well as extra caution during endoscopic surgery, may help surgeons avoid iatrogenic injury to the nerve [13,43].

In addition, there are other potential surgical mishaps. Identifying the presence of IEC on CT scans before surgical interventions, such as functional endoscopic sinus surgery, is of paramount importance. Surgeons face a significant challenge in locating the orbital floor since the base of the IEC is attached to the orbital rim. During endoscopic surgery, differentiating between the IEC wall and the orbital structure is crucial for the surgeon [44,45] to prevent orbital penetration with intra-orbital injuries [46]. Furthermore, the presence of IEC is associated with the adhesion of the orbital floor, making it more fragile, and it can cause orbital cellulitis [6].

Another concern is a partial resection of the IEC. Leaving partitions of the cell unresected can lead to complications, such as ethmoid mucocele formation [47] and osteitis [48].

### 4.4. Conclusions

Using CT scans, in addition to endoscopic tools, to detect anatomical variations near the maxillary sinuses prior to surgery is important in order to minimize iatrogenic injuries [18,46,49]. This is in line with the recommendation in the checklist before operating on the maxillary sinuses and the osteomeatal unit [18].

Surgeons should be aware of the risks of dissecting the infraorbital cell against the benefits of reducing the occurrence of sinusitis. If surgery is the best option, the surgeon should take into account that in large sinuses, there is a greater probability of damaging the infraorbital nerve (due to the greater probability of the descending type variation of IOC) [6,14,43]. Careful examination is the best surgical approach, and instrumental planning is necessary.

### 4.5. Limits of This Study and Future Thoughts

The current study has several limitations: further analysis is required to examine the complex association between maxillary sinus volume and the IEC, namely, determining whether the expansion of the sinus on the transverse plane (lateral–medial) and the presence of the IEC correlate with a wider face and whether a descending IOC correlates with the sinus and facial heights (superior–inferior). In addition, further investigation of the total pneumatization of the skull may shed light on these anatomical variations. Future studies should consider other variations of the ethmoid air cells.

## Figures and Tables

**Figure 1 diagnostics-13-03593-f001:**
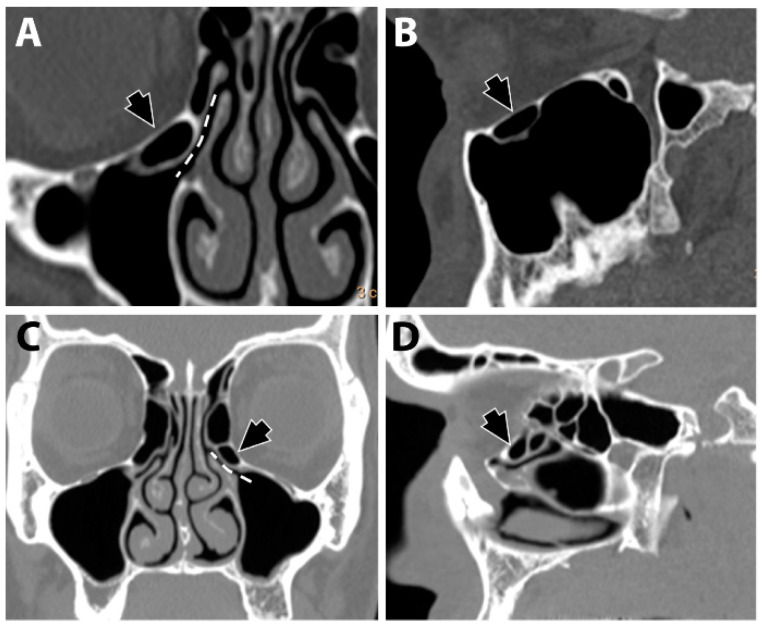
Infraorbital ethmoid cells (black arrows). A Haller cell is shown in coronal (**A**) and sagittal (**B**) views. Convoluted bulla is shown in coronal (**C**) and sagittal (**D**) views. The dashed line (**A**,**C**) denotes the location of the infundibulum.

**Figure 2 diagnostics-13-03593-f002:**
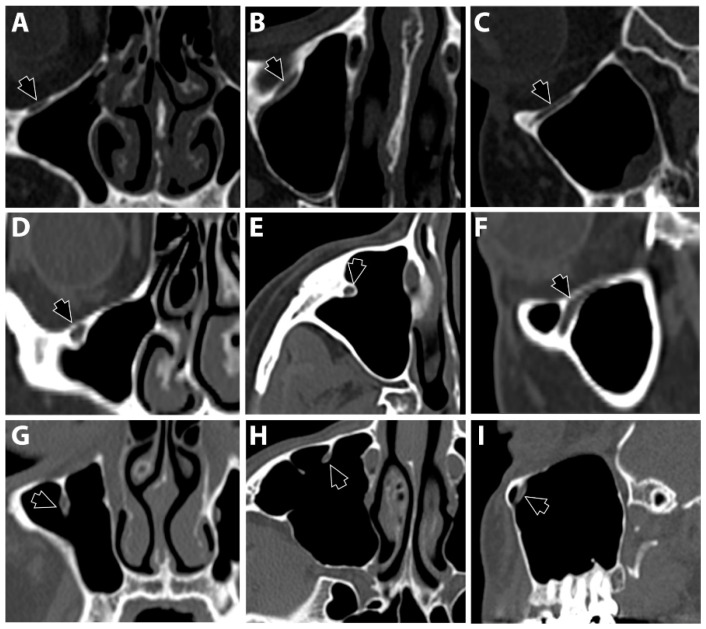
Three locations of the infraorbital canal (black arrows) following the classifications of Ference et al. and Haghnegahdar et al. [14,17] Type 1: the canal passes within the bony plate that forms the floor of the orbital cavity (**A** = coronal view, **B** = axial view, and **C** = sagittal view). Type 2: the bony canal protrudes into the lumen of the maxillary sinus (**D** = coronal view, **E** = axial view, and **F** = sagittal view). Type 3: the canal descends into the maxillary sinus (**G** = coronal view, **H** = axial view, and **I** = sagittal view).

**Figure 3 diagnostics-13-03593-f003:**
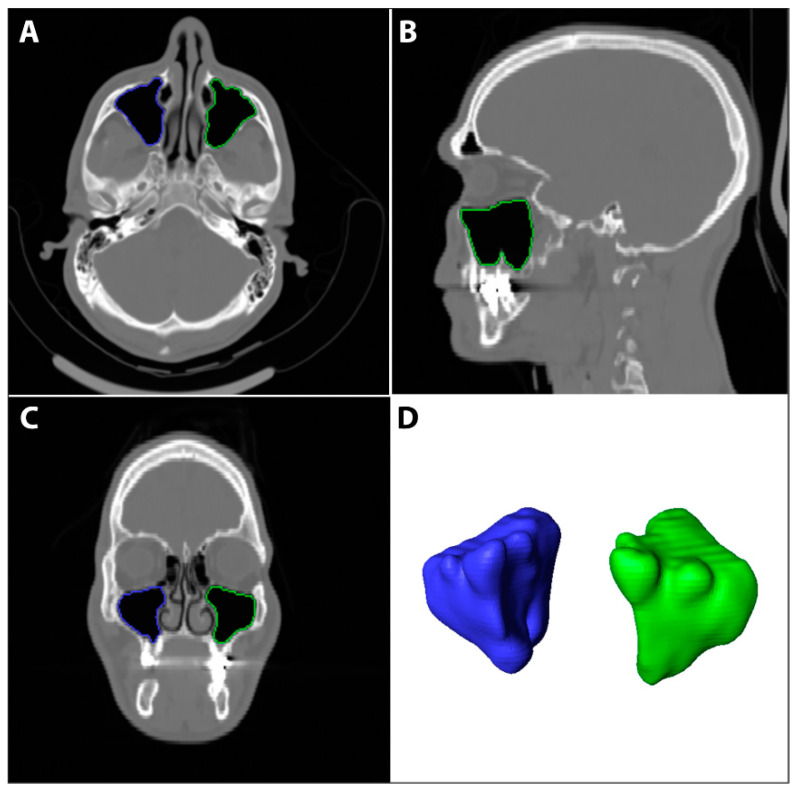
The right and left maxillary sinuses were segmented using Amira software (v. 6.3) using automated and semi-automated techniques. Using the brush and the interpolation tools, each sinus was marked on the axial (**A**), sagittal (**B**), and coronal (**C**) views and then a surface was generated from the 3D segmentation model (**D**). The volume (mm^3^) of each sinus was calculated from the surface using the ‘Surface Area Volume’ module.

**Figure 4 diagnostics-13-03593-f004:**
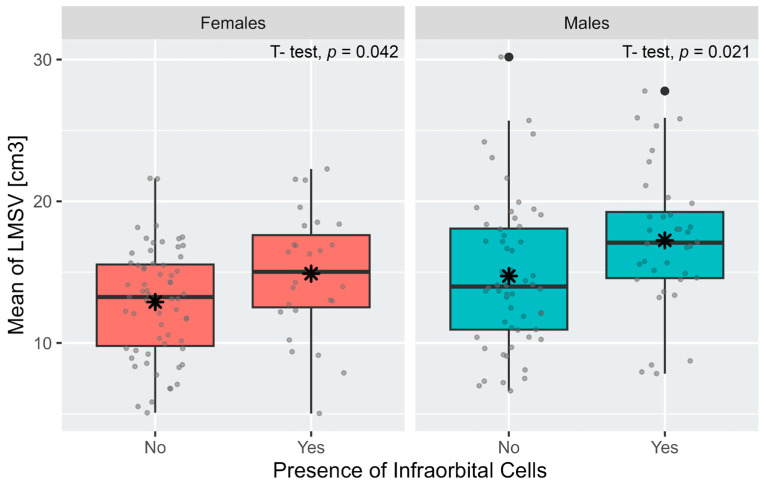
Mean (black star) and distribution (gray dots) of the left maxillary sinus volume (LMSV) in individuals with and without infraorbital ethmoid cells in males and females. For each boxplot, the box represents the first and the third quartiles, and the whiskers show Q3 + 1.5 × IQR to Q3 − 1.5 × IQR. The center line denotes the median, and the black dots denote the outliers.

**Figure 5 diagnostics-13-03593-f005:**
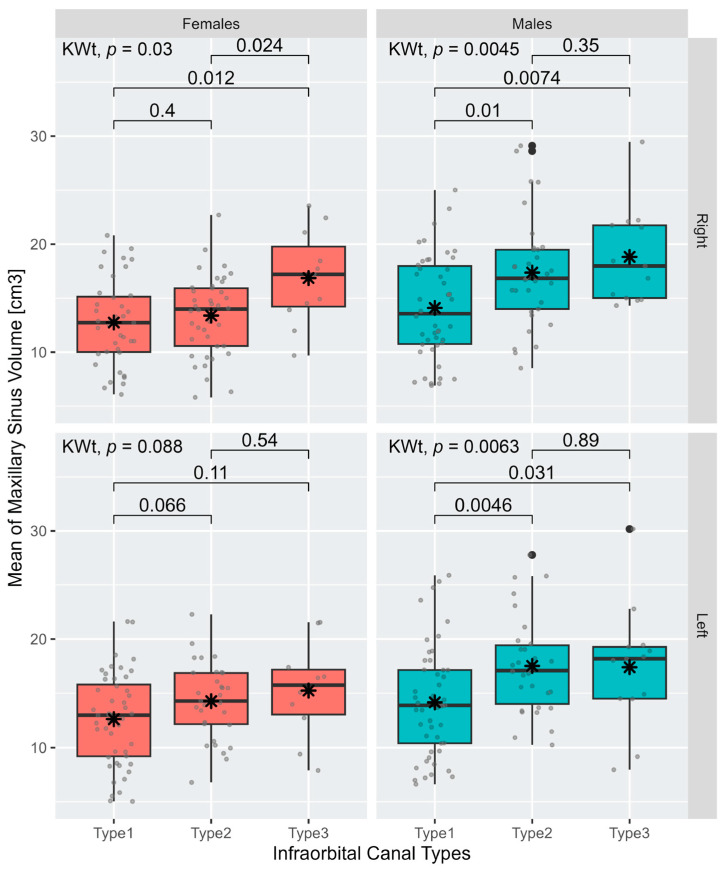
Mean (black star) and distribution (gray dots) of maxillary sinus volumes (left and right) for different types of the infraorbital canal in males and females. For each boxplot, the box represents the first and the third quartiles and the whiskers show Q3 + 1.5 × IQR to Q3 − 1.5 × IQR. The center line denotes the median, and the black dots denote the outliers. KWt: Kruskal-Wallis test.

**Table 1 diagnostics-13-03593-t001:** Prevalence of infraorbital ethmoid cells by sex (92 males and 90 females) and laterality in individuals with infraorbital ethmoid cells by sex (49 males and 31 females).

	Males *N* (%)	Females *N* (%)	*p*-Value *
Presence of IEC			
No	43 (46.7)	59 (65.6)	**0.016**
Yes	49 (53.3)	31 (34.4)	
Laterality of IEC			
Unilateral (right)	13 (26.5)	4 (12.9)	0.339
Unilateral (left)	11 (22.5)	9 (29.0)	
Bilateral	25 (51.0)	18 (58.1)	

IEC: Infraorbital ethmoid cells; * Pearson’s Chi-squared test. Significant values (*p* < 0.05) are denoted in bold.

**Table 2 diagnostics-13-03593-t002:** Prevalence of infraorbital canal types by sex (right maxillary sinus, 92 males and 90 females, Pearson’s Chi-squared test. The left side yielded similar results).

	Infraorbital Canal Types			*p*-Value
	Type-1	Type-2	Type-3	
Total *N* (%)	85 (46.7)	73 (40.1)	24 (13.2)	
Males *N* (%)	46 (50.0)	33 (35.9)	13 (14.1)	0.498
Females *N* (%)	39 (43.3)	40 (44.5)	11 (12.2)	

**Table 3 diagnostics-13-03593-t003:** Presence of infraorbital canal types by side (*N* = 182, Cochran’s Q test).

		IOC (Left) N (%)			Total-Right	*p*-Value
		Type-1	Type-2	Type-3		
IOC (right)	Type-1	78 (80.4)	6 (9.7)	1 (4.3)	85 (46.7)	0.085
	Type-2	16 (16.5)	49 (79.0)	8 (34.8)	73 (40.1)	
	Type-3	3 (3.1)	7 (11.3)	14 (60.9)	24 (13.2)	
	Total-left	97 (53.3)	62 (34.1)	23 (12.6)	182 (100.0)	

IOC: Infra orbital canal.

**Table 4 diagnostics-13-03593-t004:** Logistic regression coefficients that determine the factors affecting the probability of having infraorbital cells (right maxillary sinus, *N* = 182. The left side yielded similar results).

	Estimate	Std. Error	z Value	Pr (>|z|)
(Intercept)	−2.7361	0.5895	−4.641	**<0.001**
Sex Male	0.5307	0.3403	1.560	0.119
RMSV	0.1149	0.0370	3.105	**<0.01**

RMSV: right maxillary sinus volume. Significant values (*p* < 0.05) are denoted in bold.

**Table 5 diagnostics-13-03593-t005:** Association between the infraorbital canal types and the presence of infraorbital ethmoid cells (the right maxillary sinus, Pearson’s Chi-squared test).

		Infraorbital Canal Type			*p*-Value
		Type-1	Type-2	Type-3	
**IEC** *N* (%)	No (*n* = 122)	69 (56.6)	43 (35.2)	10 (8.2)	**<0.001**
	Yes (*n* = 60)	16 (26.7)	30 (50.0)	14 (23.3)	

IEC: Infraorbital ethmoid cells. Significant values (*p* < 0.05) are denoted in bold.

**Table 6 diagnostics-13-03593-t006:** Multiple logistic regression coefficients were used to determine factors that affected the probability of having a type 3 infraorbital nerve canal (the right maxillary sinus, *N* = 182).

	Estimate	Std. Error	z Value	*p* (>|z|)
(Intercept)	−4.406164	1.138936	−3.869	**<0.001**
Sex Male	−0.447697	0.498864	−0.897	0.370
IEC	0.828927	1.760585	0.471	0.638
RMSV	0.145169	0.069290	2.095	**<0.005**
IEC:RMSV	0.008495	0.098510	0.086	0.931

IEC: infraorbital ethmoid cells; RMSV: right maxillary sinus volume. Significant values (*p* < 0.05) are denoted in bold.

**Table 7 diagnostics-13-03593-t007:** Associations between the recorded sinusitis and the presence of infraorbital cells or infraorbital canal types.

Anatomic Variation		Recorded Sinusitis		Pearson’s Chi-Squared Test
		No	Yes	
IEC	No	79 (56.8)	23 (53.5)	χ^2^ = 0.044, df = 1, *p* = 0.833
	Yes	60 (43.2)	20 (46.5)	
IOC (right)	Type 1	76 (54.7)	21 (48.8)	χ^2^ = 3.330, df = 2, *p* = 0.189
	Type 2	43 (30.9)	19 (44.2)	
	Type 3	20 (14.4)	3 (7.0)	
IOC (left)	Type 1	65 (46.8)	20 (46.5)	χ^2^ = 0.030, df = 2, *p* = 0.985
	Type 2	56 (40.3)	17 (39.5)	
	Type 3	18 (12.9)	6 (14.0)	

IEC: infraorbital ethmoid cells; IOC: infraorbital canal type.

## Data Availability

The data presented in this study are available on request from the corresponding author.

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
