# Peer review of "The Associations between the Maxillary Sinus Volume, Infraorbital Ethmoid Cells, and the Infraorbital Canal: A CT-Based Study"

_diagnostics, 2023, doi:10.3390/diagnostics13233593_

Round 1
Reviewer 1 Report
Comments and Suggestions for Authors
It seems like a good paper that clearly describes anatomical variants using precise terminology and accurate statistical analysis.
It may be advisable to consider removing the content on 'Sinusitis' in the Results section, as the information provided is insufficient either in the Results or the Discussion.
Finally, the conclusion section appears to be inadequately written. Please provide additional details.
The excessive number of tables and figures is affecting readability. It could be worth considering moving some of the detailed analysis tables and figures to the supplementary material.
Author Response
- Summary
Thank you very much for taking the time to review this manuscript. Please find the detailed responses below.
- Questions for General Evaluation
Are the results clearly presented?
Reviewer’s Evaluation: Can be improved
Response and Revisions: Improved as requested
Are the conclusions supported by the results?
Reviewer’s Evaluation: Must be improved
Response and Revisions: Improved as requested
- Point-by-point response to Comments and Suggestions for Authors
It seems like a good paper that clearly describes anatomical variants using precise terminology and accurate statistical analysis.
Comments 1: It may be advisable to consider removing the content on 'Sinusitis' in the Results section, as the information provided is insufficient either in the Results or the Discussion.
Response 1: Several studies have found that sinusitis correlates with IEC. This kind of information contributes to the treatment decision. We wanted to determine whether the association appears in our sample as well, and found no correlation. We believe that the data we provide regarding sinusitis and IEC are important for otolaryngologists to decide on the right treatment for chronic sinusitis, and to avoid unnecessary surgical procedures. This is especially important when considering the benefits against the risk of removing the infraorbital cells. We revised the clinical implication’s section in the discussion to clarify this point.
Comments 2: Finally, the conclusion section appears to be inadequately written. Please provide additional details.
Response 2: Agree. We have separated the conclusion section (4.4) from the rest of the text and added more details as requested.
Comments 3: The excessive number of tables and figures is affecting readability. It could be worth considering moving some of the detailed analysis tables and figures to the supplementary material.
Response 3: We could not make changes in the figures since each figure reflects the essence of the article. However, we made some technical changes in the tables in order to reduce their number (from 9 to 7), without changing the data in the results. The following changes were carried out:
Tables 1 and 2 were combined to a single table (new Table 1).
Table 8 was removed. The data are now included in the results section.
All other tables were renumbered according to these changes.
It was important for us to show the complex relationships between IEC, IOC, and the sinus size volume.
Reviewer 2 Report
Comments and Suggestions for Authors
Dear authors
Thank you for the opportunity to get acquainted with your interesting work.
The work requires corrections.
In the introduction, the authors should include information about other differences of the ethmoid sinuses and supplement the literature DOI: 10.1016/j.ijporl.2017.04.001 and https://doi.org/10.1002/ca.24109
The authors should supplement it in the material methodology with clear inclusion and exclusion criteria in the study group.
Include limitations of the work in the discussion.
Comments on the Quality of English LanguageThere are many spelling and language mistakes and the manuscript needs to be corrected by a native English speaker.
Author Response
- Summary
Thank you very much for taking the time to review this manuscript. Please find the detailed responses below.
- Questions for General Evaluation
Does the introduction provide sufficient background and include all relevant references?
Reviewer’s Evaluation: Must be improved
Response and Revisions: Improved as requested
Are all the cited references relevant to the research?
Reviewer’s Evaluation: Can be improved
Response and Revisions: Improved as requested
Is the research design appropriate?
Reviewer’s Evaluation: Must be improved
Response and Revisions: Improved as requested
- Point-by-point response to Comments and Suggestions for Authors
Thank you for the opportunity to get acquainted with your interesting work.
The work requires corrections.
Comments 1: In the introduction, the authors should include information about other differences of the ethmoid sinuses and supplement the literature DOI: 10.1016/j.ijporl.2017.04.001 and https://doi.org/10.1002/ca.24109
Response 1: Agree. There are indeed many anatomical variations in the paranasal sinuses and nasal cavity. The authors are aware of the various types of air cells such as the Onodi cell and the agger nasi. As requested, we added this information (including the appropriate references) to the introduction section. The current study focuses on the air cell variation next to the maxillary sinuses (ION and IEC); this is why we did not refer to the other cells mentioned in the introduction in the discussion section.
Comments 2: The authors should supplement it in the material methodology with clear inclusion and exclusion criteria in the study group.
Response 2: Agree. In the revised paper, we expended the inclusion and exclusion criteria, as requested. Moreover, in the limitation section we recommended that other variations of the ethmoidal air cells should be considered with regard to the maxillary sinus volume.
Comments 3: Include limitations of the work in the discussion.
Response 3: Agree. The limitations of the study are described in section 4.5
Comments 3: Comments on the Quality of English Language- There are many spelling and language mistakes and the manuscript needs to be corrected by a native English speaker.
Response 3: Thank you for pointing this out. The paper was edited by a scientific editor who is a native English speaker. Nevertheless, the revised manuscript was again reviewed by a native English speaker.
Reviewer 3 Report
Comments and Suggestions for Authors
This article is well thought out and written. I think the sample size is adequate. There are no objections to the content . If I may venture to say so, the significance of this study could be emphasized more, as mentioned in Discussion 4.3 Clinical Implications. A more detailed explanation, such as whether the presence or absence of IEC increases the difficulty of the procedure, would help the reader better understand the clinical utility of this study. As the authors state in 4.4 Limits, further studies are expected in the future.
Author Response
Thank you very much for taking the time to review this manuscript. Please find the detailed responses below.
Point-by-point response to Comments and Suggestions for Authors
This article is well thought out and written. I think the sample size is adequate. There are no objections to the content.
Comments 1: If I may venture to say so, the significance of this study could be emphasized more, as mentioned in Discussion 4.3 Clinical Implications. A more detailed explanation, such as whether the presence or absence of IEC increases the difficulty of the procedure, would help the reader better understand the clinical utility of this study.
Response 1: Agree, thank you for this comment. The clinical implications section (4.3) was revised and a more detailed explanation was added as requested.
Comments 2: As the authors state in 4.4 Limits, further studies are expected in the future.
Response 2: Agree.
Round 2
Reviewer 2 Report
Comments and Suggestions for Authors
Dear authors
Thank you for the corrections you made
I accept in present form